# Cohort profile: a longitudinal study of HIV infection in the central nervous system with focus on cerebrospinal fluid – the Gothenburg HIV CSF Study Cohort

Lars Hagberg  ,[1,2] Magnus Gisslén[1,2]

¹Department of Infectious Diseases, Institute of Biomedicine, University of Gothenburg Sahlgrenska Academy, Gothenburg, Sweden
²Department of Infectious Diseases, Region Västra Götaland, Sahlgrenska University Hospital, Gothenburg, Sweden

**Correspondence to**
Dr Lars Hagberg;
lars.hagberg@gu.se

## ABSTRACT

**Purpose** In order to enable long-term follow-up of the natural course of HIV infection in the central nervous system, a longitudinal cohort study with repeated cerebrospinal fluid (CSF) analyses at intervals over time was initiated in 1985. When antiretrovirals against HIV were introduced in the late 1980s, short-term and long-term effects of various antiretroviral treatment (ART) regimens were added to the study.

**Participants** All adult people living with HIV (PLWH) who were diagnosed at or referred to the Department of Infectious Diseases, Sahlgrenska University Hospital, Gothenburg, Sweden were asked to participate in the Gothenburg HIV CSF Study Cohort. PLWH with neurological symptoms or other clinical symptoms of HIV, as well as those with no symptoms of HIV infection, were included. Most participants were asymptomatic, which distinguishes this cohort from most other international HIV CSF studies. In addition, HIV-negative controls were recruited. These included people on HIV pre-exposure prophylaxis who served as lifestyle-matched controls to HIV-infected men who have sex with men. Since lumbar puncture (LP) is an invasive procedure, some PLHW only consented to participate in one examination. Furthermore, at the beginning of the study, several participants were lost to follow-up having died from AIDS. Of 662 PLWH where an initial LP was done, 415 agreed to continue with follow-up. Among the 415, 56 only gave permission to be followed with LP for less than 1 year, mainly to analyse the short-term effect of ART. The remaining 359 PLWH were followed up with repeated LP for periods ranging from >1 to 30 years. This group was defined as the 'longitudinal cohort'. So far, on 7 April 2022, 2650 LP and samplings of paired CSF/blood had been performed, providing a unique biobank.

**Findings to date** A general finding during the 37-year study period was that HIV infection in the central nervous system, as mirrored by CSF findings, appears early in the infectious course of the disease and progresses slowly in the vast majority of untreated PLWH. Combination ART has been highly effective in reducing CSF viral counts, inflammation and markers of neural damage. Minor CSF signs of long-term sequels or residual inflammatory activity and CSF escape (viral CSF blips) have been observed during follow-up. The future

## STRENGTHS AND LIMITATIONS OF THIS STUDY

⇒ Strength of our study is its uniquely long follow-up time, with cerebrospinal fluid (CSF) data from a population of people living with HIV with a predominately neuroasymptomatic clinical appearance.
⇒ A strict protocol for collecting and storing CSF/blood samples at one centre and only engaging a handful of clinicians to enhance consistency and uniformity is another strength.
⇒ Difficult to follow the protocol with yearly repeated lumbar puncture in several participants due to consent and COVID-19 pandemic is a limitation.
⇒ Infrequent use of extensive neuropsychiatric testing.
⇒ Only limited number of participants suffering from severe neurological complications and opportunistic central nervous system infections.

course of these changes and their clinical impact require further studies.

**Future plans** PLWH today have a life expectancy close to that of non-infected people. Therefore, our cohort provides a unique opportunity to study the long-term effects of HIV infection in the central nervous system and the impact of ART and is an ongoing study.

## INTRODUCTION

Chronic untreated HIV infection causes progressive immunodeficiency that leads to AIDS in a median of 10 or 11 years after a primary infection. Before effective treatment was available, HIV-associated dementia was frequently observed in the late stages of the infection. The introduction of combination antiretroviral treatment (ART) that preserves or restores immune functions has had a major impact on morbidity and mortality. It has also resulted in a marked reduction of HIV-associated dementia and other neurological complications.[1] Nevertheless, mild forms of HIV-associated neurocognitive disorders have frequently been reported even during

suppressive ART.[2] The research questions for the present study have been: can biomarkers in cerebrospinal fluid (CSF) be used to chart the natural course of central nervous system (CNS) HIV infection and can we determine the short-term and long-term effects of ART?

## COHORT DESCRIPTION AND METHOD
### Study population
The study was initiated in 1985 when two patients attending our clinic presented with Guillain Barre syndrome as a complication to HIV infection and exhibited HIV isolated in the CSF.[3] Later we found that HIV-1 could also be isolated in CSF of virus carriers who had no neurological symptoms.[4] At that time, it was not known that HIV was neurotropic.

Since 1985, HIV-infected people living with HIV (PLWH) in the Gothenburg area of Sweden have been enrolled in a longitudinal study with serial sampling of CSF and blood. Lumbar punctures (LPs) are performed annually, if possible, or more frequently in connection with introduction or cessation of ART. Both PLWH with neurological or other clinical symptoms of HIV and asymptomatic PLWH have been included in the study.

Of 662 PLWH in whom an initial LP was done, 415 PLWH agreed to continue with follow-up. Among the 415, 56 only gave permission to be followed with LP for less than 1 year, mainly to analyse the short-term effect of ART. The remaining 359 PLWH were followed with repeated LP for periods >1 up to 30 years. This group was defined as the 'longitudinal cohort' (see flow chart in figure 1). The 247 PLWH who participated with only one CSF/blood sampling have been included in many cross-sectional studies. In this ongoing cohort study, we present data from 1985 to 7 April 2022 including 2650 paired CSF/blood samples, providing a unique and continuously growing biobank.

### Longitudinal cohort
In total, 359 PLWH were followed with a mean number of 6.26 (range 2–30) LP and CSF/blood analyses over a mean period of 6.89 years (range 1–30 years). Patient characteristics, time of follow-up and number of CSF analyses are shown in table 1, and time for inclusion and number of PLWH still eligible for follow-up are shown in table 2.

### Patient and public involvement
Development of the research question, outcome measures and presentation of results has been done with the local PLWH organisation PG Vast.

About half of the participants (51%) were born in Sweden (n=182) and 177 (49%) outside Sweden, most commonly in Sub-Saharan Africa (n=93; 26%), Europe outside Sweden (n=37; 10%) and Asia (n=35; 10%). The COVID-19 pandemic partially halted the study for a period. However, 166 PLWH (46%) had been followed for more than 5 years, and 65 of these (18%) have been followed for more than 10 years, providing great opportunities for longitudinal evaluations. A total of 121 PLWH were diseased or lost to follow-up. The remaining 238 PLWH are eligible to ask for further follow-up (see flow chart in figure 1).

In addition, 94 HIV-negative healthy controls have been recruited for CSF/blood sampling, of whom 53 are men on HIV pre-exposure prophylaxis (PrEP). Follow-up sampling of this cohort is ongoing.

### Validation of the study population
PLWH whose CSF/blood was sampled only once and declined to continue the study because of discomfort from the initial LP, were too sick for follow-up, were below 18 years of age or were lost to follow-up were registered in the reject log. That log included 247 PLWH with a mean age of 40.1 years (range 2–76), of whom 74 (30%) were

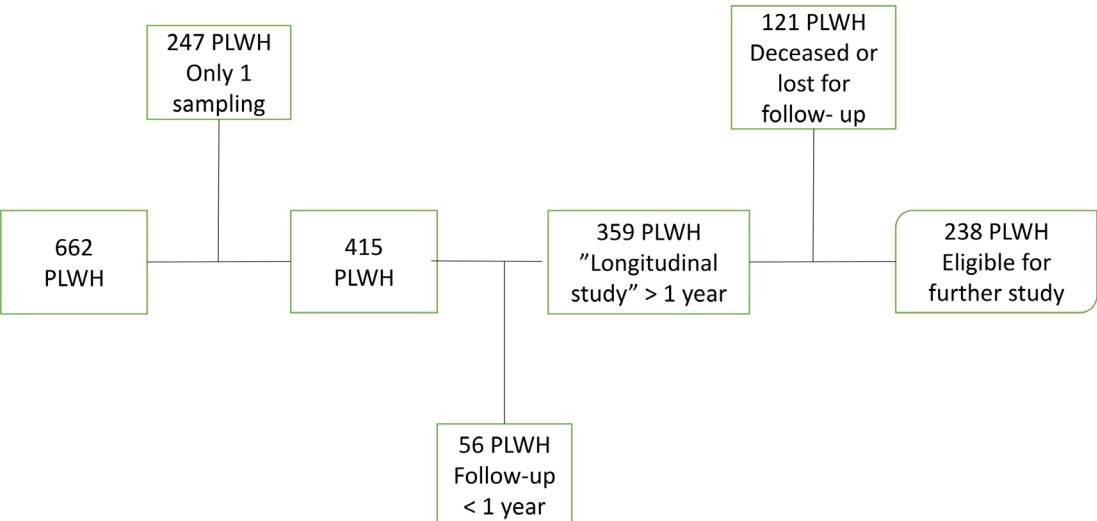

**Figure 1** Flow chart of 662 people living with HIV (PLWH) included in the Gothenburg HIV cerebrospinal fluid study (1985 to 7 April 2022).

**Table 1** Longitudinal cohort characterisation

| | |
|---|---|
| Number of participants followed >1 year | 359 |
| Mean time of follow-up | 6.89 years (range 1–30) |
| Number of lumbar punctures/individuals | 6.26 (range 2–30) |
| Age at inclusion, mean | 40.7 years (range 17–73) |
| Gender, men/women | 247/112 |
| Geographic background (n) | Sweden 182 |
| | Europe (outside Sweden) 37 |
| | Africa 93 |
| | Asia 35 |
| | Middle East 3 |
| | America 9 |
| HIV1/HIV2 | 356/3 |
| Mean CD4$^+$ cells at inclusion | 369×10$^6$/L (range 0–2131) |
| CDC classification* at inclusion | A1–A3 n=244; B1–B3 n=34; C1–C3 (AIDS) n=81 |
| Antiretroviral treatment at inclusion (n) | No treatment: 298 (of whom 31 had primary infection) |
| | Treatment suppressed: 49 |
| | Treatment failure: 11 |
| | Treatment interruption: 1 |
| Neurology at inclusion (n) | Neuroasymptomatic: 328 |
| | CNS opportunist: 12 |
| | HIV-associated dementia: 10 |
| | Other CNS complications: 9 |

*Revised classification system for HIV: MMWR Dec 18; 41:1–19. 1993.
CNS, central nervous system.

women and 173 (70%) were men. Their mean CD4 cell count was 344×10$^6$/L (range 10–1420); 40% were classified as CDC stage C1–C3 (AIDS). The number of AIDS patients (40%) were larger than the longitudinal cohort

**Table 2** Time for inclusion (5 years interval) in the cohort study, number of LP:s and number of PLWH still eligible for follow-up

| PLWH no | LP no (range) | PLWH still eligible no |
|---|---|---|
| 1985–1990 | 36 | 247 (2–20) | 5 |
| 1990–1995 | 32 | 216 (2–30) | 7 |
| 1995–2000 | 54 | 442 (2–27) | 22 |
| 2000–2005 | 48 | 396 (2–19) | 33 |
| 2005–2010 | 48 | 328 (2–19) | 40 |
| 2010–2015 | 62 | 326 (2–16) | 57 |
| 2015–2020 | 74 | 292 (2–7) | 67 |

LP, lumber puncture; PLWH, people living with HIV.

group (22%), but otherwise age, gender and CD4 cell count of similar magnitude between the groups.

## Cohort variables and laboratory analyses

Full clinical history, including comorbidity, treatment and laboratory results, was collected throughout the study. Variables recorded at enrolment included sex, country of birth, mode of HIV transmission, date of last HIV-negative test (if any), and first positive HIV-test and suspected country for HIV transmission. Data collected/updated at each follow-up visit included ART, prophylaxis of opportunistic infections and comedications administered. Data on coinfection with hepatitis C and B virus, weight, date and type of AIDS-defining events and non-AIDS events, and date and cause of death were also included. HIV-RNA, CD4+ and CD8+ T cell counts and CD4/CD8 ratios, HIV subtype and HIV drug resistance (including viral sequences) were also recorded.

CSF and blood collection took place in the morning, before breakfast in a standardised manner with the subject in a lateral recumbent position. Twenty-four mL of CSF were collected and centrifuged for cell counting. CSF cells and buffy coat were stored separately, and cell-free CSF, serum and plasma were divided into fractions. Fractions not immediately analysed were stored at −70°C in the local laboratory after collection. In conjunction with each LP, virological, immunological and neuronal injury markers in the CSF were compared with the clinical course. The laboratory analyses were grouped in three categories.

1. Virology.
2. Markers of inflammation and immunology.
3. Markers of CNS injury.

1. *Virology*: Since its introduction in 1996, quantitative HIV-1 RNA PCR has been used as the main marker of viral load (currently Cobas Taqman v.2, Roche Diagnostic Systems, Hoffmann-La Roche, Basel, Switzerland). Quantitative HIV-1 DNA real-time PCR (TaqMan5' nuclease) assay has been analysed in subpopulations. Prior to 1996, CSF HIV antigen test and virus isolation were included in the protocol.

2. *Markers of inflammation and immunology*: Neopterin concentration reflects macrophage activity and has been the main marker of CSF inflammation (RIA method Henning test Neopterin, BRAHMS, Berlin, Germany). Among other procedures, CSF monocyte cell count, protein electrophoresis, oligoclonal bands and beta-2-microglobulin concentration have been analysed in all patients. Various cytokines such as MCP-1, IP-10, CXCL10, uPA and suPAR have also been measured in subpopulations.

3. *Markers of CNS injury*: CSF neurofilament light protein (Nfl) concentration that reflects axonal damage has been the main marker used to estimate CNS injury in CSF (NF-light ELISA kit; Uman Diagnostics AB, Umeå, Sweden), and ultrasensitive single molecule array (Simoa) method has been used for blood. Other markers such as gangliosides (GM1, GD1a, GD1b and

GT1b), sulfatides and glial markers, including GFAP (astroglia) and GD3 (microglial/macrophages), t-tau, p-tau, beta-amyloids, s-APP and neurogranin (synapse protein), have been studied in subpopulations.

### Neuropsychiatric test

Reaction time tests have been performed in subpopulations.[56] Since 2011, neurocognitive testing has been done with a computerised cognitive test battery (Cogstate, Melbourne, Australia) that has been validated for HIV-infected individuals.[7 8] Four different tests from the Cogstate Brief Battery were used to assess five cognitive domains: the Detection Test measured psychomotor function and attention, the Identification test assessed speed of information processing and attention, the One Card Learning test evaluated learning and the One Back memory test assessed working memory.[9]

### Statistical methods

In most studies, Wilcoxon signed rank test was used to compare the variables before and after treatment. Differences between groups were assessed with the Mann-Whitney U test.

### Collaboration

Batches of CSF and serum samples are stored in −70°C in PLWH with clinical and laboratory data described above and will be available for potential collaborators studying biomarkers during HIV infection.

### RESULTS

A general finding resulting from our 37-year study is that HIV-1 infection in the CNS, as mirrored by CSF findings, appears early during the infectious course of the disease and progresses slowly in the vast majority of untreated PLWH.[10 11] Combination ART has been very effective in reducing CSF HIV-1 loads, inflammation and markers of neural damage.[12 13] However, minor CSF signs of long-term sequela or residual inflammatory activity have been observed during follow-up.[14–17]

When several CSF specimens from the same individual were studied, HIV-1 could be isolated from 80% of them and detected by PCR in 90% of cases,[10] with higher HIV-RNA cut-off levels required in CSF than in blood to predict positive HIV-1 isolation.[18] The viral load was approximately (mean) 1 log lower in CSF than in blood,[19–21] but CSF viral load exceeded plasma levels (CSF>plasma discordance) in 13%, with variations between different disease stages, ranging from 1% in primary HIV, 11% in neuroasymptomatic patients and up to 30% in patients with HIV-associated dementia.[19 21] HIV-1 RNA levels increased in CSF relative to time of infection.[11] Markers of immune stimulation such as neopterin and beta-2 microglobulin also increased in CSF during follow-up, indicating that HIV-1 CNS infection is progressive, even in a neurological asymptomatic stage.[22] However, CSF

pleocytosis decreased in PLWH with severe immunosuppression,[15 19 21] probably because of T cell deficiency.

Monotherapy ART with zidovudine, the only existing drug at the beginning of the AIDS epidemic, resulted in a 53%–57% decrease in CSF neopterin concentrations.[23] The next drug on the market, didanosine, had no such effect.[24] Zidovudine-resistant variants in the brain developed during monotherapy.[25] It was obvious that combination treatment was necessary to avoid viral resistance, and in 1996, when protease inhibitors was added to two nucleoside reverse transcriptase inhibitors, a breakthrough in HIV care was achieved. Blood HIV load decreased to numbers below 50 copies/mL in most PLWH, and those with HIV-related symptoms clinically improved.

We found that ART was very effective in reducing viral load in CSF. Ultrasensitive PCR showed that highly active ART (HAART) resulted in undetectable HIV-1 RNA copies in CSF, although the virus was still detectable in plasma.[12 26] Several ART combinations proved virologically effective in CSF[26–29] and reduced blood-brain barrier integrity and intrathecal immunoglobulin production.[30 31] CSF markers of axonal injury were also normalised.[13 17 28 32]

Studies of the highly effective short-term effects of ART were followed by long-term studies. We found that CSF viral loads were effectively suppressed over long periods of observation, but CSF signs of slight immune activation were still present in many PLWH after several years of suppressive treatment.[14 16 33]

While CNS infection is generally well controlled by systemic suppressive ART, there are exceptions when the HIV-RNA load increases in the CSF despite suppression of the plasma viral load, a phenomenon referred to as asymptomatic CSF viral escape.[34] In these PLWH, CSF viral counts often reach just above 50 copies/mL without accompanying CSF pleocytosis or CSF signs of neuronal injury, measured as increased CSF concentration of Nfl. The lack of clinical symptoms, and the fact that the viral CSF increase was most often transient and reversed without changing therapy, has resulted in this condition to be interpreted as benign 'CSF viral blips', similar to plasma blips.[35] CSF escape is associated with increased CSF neopterin concentrations and may be related to the size of the CNS HIV reservoir. Correspondingly, residual CSF viral loads below the limit of quantification by standard assays also correlate with the degree of CSF immune activation in PLWH receiving suppressive ART.[26 36] This reinforces the view that intrathecal immune activation is driven by persistent virus in the CNS. Nevertheless, similar to findings during systemic infection,[37] treatment intensification does not seem to decrease the residual CSF viral load or inflammation,[38 39] suggesting that there is no ongoing HIV replication during effective treatment.

Although it has not yet been definitely proven, data suggest that a stable permanent infection of cells in the CNS is established later than in the systemic viral reservoirs,[40 41] which are highly concentrated in memory T cell compartments within the first days of systemic HIV

infection.[42] When examining anti-HIV antibody formation as a surrogate marker for antigen load and the size of the viral reservoir[43] in patients followed longitudinally during early HIV infection, serum anti-HIV antibodies emerged in blood by day 30 in untreated early infection, while CSF antibodies reached similar levels about 2 weeks later.[44] In addition, high-antibody levels, comparable to those observed in chronically infected subjects, were reached several months later in CSF, as compared with blood. In addition, while treatment of chronic infection resulted in only small reductions in levels of anti-HIV antibodies in both CSF and serum, treatment during early infection substantially reduced CSF levels of antibodies, sometimes to levels close to those in HIV-negative controls. In contrast, antibody levels in serum were less affected,[44] altogether supporting this hypothesis. Our findings and those from other groups further support compartmentalisation of HIV infection and immune activation in the CNS.[45–48]

The low-grade CNS immune activation found during suppressive ART may not be solely ascribed to HIV itself since comorbidities, coinfections and lifestyle-related factors can contribute, as elegantly shown in the COBRA study.[49 50] The importance of appropriate controls was also demonstrated in our HIV-negative PrEP controls in whom immune activation markers and signs of neuronal injury increased as compared with non-PrEP controls.[51]

Occasionally, CSF viral escape, that is, viral load in CSF but not in blood, is accompanied by HIV-induced neurological and neurocognitive signs and symptoms, which have been defined as 'symptomatic CSF escape'.[52–54] Another phenomenon that causes increased CSF viral load in well-treated PLWH is a concomitant infection in the nervous system. As an example, herpes zoster sometimes results in an inflammatory CSF reaction with slight pleocytosis, increased CSF neopterin concentrations and increased CSF viral load, but with no detectable plasma virus. This phenomenon has been called 'secondary CSF escape' and may be the result of latent virus released or detected from activated monocytes.[55]

## DISCUSSION

HIV-1 infects many compartments in the human body, including the CNS. CSF surrounds the brain and is a fluid accessible to LP, which can give valuable information on infectious activity and pathological processes in the CNS. As noted earlier, in our clinical cohort, most participants were asymptomatic. It can be a challenge to enrol PLWH to do repeated LP. For this reason, most cohorts studying CSF only include participants with neurological and cognitive complications or opportunistic CNS infections having limited follow-up. To survey the whole panorama of the infectious course, we have a long-lasting collaboration with other centres, the most important being cohorts at UCSF, San Francisco, California, USA and Milan, Italy. This enables us to compare our cohort with

a large number of PLWH suffering from AIDS dementia complex and other CNS complications.

HIV infection changed dramatically in severity since we began this study in 1985. At that time, we had no idea that a combination of antiretroviral drugs could have such impact. The natural course of the infection was observed with considerable data showing progressive CNS disease in several PLWH. When ART was introduced, the longitudinal project changed to monitor whether CSF viral load, markers of inflammation and CNS injury became normalised. Some early medication with monotherapy had a limited short-term effect. It was not until 1996 when HAART with three drugs begun that the disease became a chronic latent infection with a long-life expectancy and a high quality of life. However, PLWH must remain life-long on ART. Luckily, up to now there are many modern drugs to choose from with no or minor adverse events.

There has been a debate over whether the most frequently used criteria for cognitive impairment in people with HIV, namely the Frascati criteria developed in 2007,[56] overestimate cognitive impairment. New criteria that are more appropriate for the modern era have been sought.[57 58] The significance of mild forms of neurocognitive disorders and asymptomatic cognitive impairment detected in well-treated PLWH is controversial. Furthermore, if this condition exists, are they progressive or reversible?[2 59] Increased CSF inflammation has been reported in PLWH on suppressive ART who experience mild cognitive impairment,[60] but the implications of this are yet to be settled. The role of ageing, underlying diseases and lifestyle factors of mild neurocognitive disorder are largely unknown. In such patients, longitudinal CSF studies are helpful to determine pathogenic factors that may affect the CNS.

Despite our intention to follow participants annually, it was difficult to accomplish. Some participants consented to repeated LP but at longer intervals than annually. Moreover, the COVID-19 pandemic halted several clinical studies for 2 years. Another limitation is that our cohort included relatively few PLWH with advanced disease and CNS complications. By means of international collaborations, more CSF data from patients with severe neurological complications and opportunistic CNS infections have been used for comparison with our cohort data in several cross-sectional studies. A limitation is also the infrequent use of extensive neuropsychiatric test batteries. Several attempts to include regular neuropsychiatric tests in the protocol failed due to methodological difficulties and a lack of resources. Furthermore, it was difficult to enrol a valid control group. Another complication while managing neuropsychiatric analyses in longitudinal studies involving repeated tests is the learning factor, which may result in false test results. In addition, there were many participants with language difficulties.

A major strength of our study is its uniquely long follow-up time, with CSF data from a population of PLWH with a predominately neuroasymptomatic clinical appearance, which we believe has never been done before.

In addition, the study was performed with a very strict protocol for collecting and storing CSF/blood samples at one centre and only engaging a handful of clinicians to enhance consistency and uniformity.

Our future objectives involve several questions. Are there any active, ongoing inflammatory or neurotoxic processes remaining in the brain despite successful virological treatment? Are there any complications from the CNS caused by chronic antiretroviral medication? Characterisation of the CNS HIV reservoir and its establishment is still largely unknown. What is the importance of compartmentalised CNS infection, and if peripheral eradication treatment in the blood and the lymphatic system proves successful in the future, is it possible to bring about the same effect in the CNS? The study will continue with follow-up of already included participants and recruitment of newly diagnosed PLWH.

**Contributors** LH and MG designed the study, examined and collected CSF from the majority of the participants, registered the results and wrote the paper with equal contribution. LH is responsible for the overall content as the guarantor.

**Funding** This work was supported by the Swedish state under an agreement between the Swedish government and the county councils (ALF agreement ALFGBG-965885) and the Swedish Research Council (2021-06545).

**Competing interests** LH has no competing interest. MG has received research grants from Gilead Sciences and Janssen-Cilag, and honoraria as a speaker and/or scientific advisor from Amgen, AstraZeneca, Biogen, Bristol-Myers Squibb, Gilead Sciences, GlaxoSmithKline/ViiV, Janssen-Cilag, MSD, Novocure, Novo Nordic, Pfizer and Sanofi.

**Patient and public involvement** Patients and/or the public were involved in the design, conduct, reporting or dissemination plans of this research. Refer to the Cohort description and method section for further details.

**Patient consent for publication** Not applicable.

**Ethics approval** This study involves human participants and was approved by the Regional Ethics Review Board in Gothenburg, Sweden (Ö588-01). Participants gave informed consent to participate in the study before taking part.

**Provenance and peer review** Not commissioned; externally peer reviewed.

**Data availability statement** Data are available upon reasonable request.

**ORCID iD**
Lars Hagberg http://orcid.org/0000-0002-5685-297X

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
