## [Reviewer comments · BMJ Open]

ARTICLE DETAILS

TITLE (PROVISIONAL)	Cohort profile: A longitudinal study of HIV infection in the central nervous system with focus on cerebrospinal fluid-The Gothenburg HIV CSF Study Cohort
AUTHORS	Hagberg, Lars; Gisslén, Magnus

VERSION 1 – REVIEW

REVIEWER	Azevedo , Beatriz Universidade Federal do Rio de Janeiro
REVIEW RETURNED	22-Jan-2023

GENERAL COMMENTS	First of all, congratulations for the work developed by the authors, the compilation of information and findings are extremely important in the development of studies on HIV, the Central Nervous System and its correlation with antiretroviral therapies. I only make one observation, I did not visualize the statistical part of the work, if any test or statistical program was carried out.
--

REVIEWER	Shi, Yuxin Fudan University
REVIEW RETURNED	27-Feb-2023

GENERAL COMMENTS	The Gothenburg HIV CSF Study Cohort is a valuable cohort in studying the HIV effect on central nervous system (CNS). The authors have demonstrated the natural course of HIV infection and the impact of ART within CNS in their previous publications using data from this long-term follow-up study. I have a few suggestions. Major 1. More details of the cohort or the subgroups are needed. Figure 1 provides a nice summary of the sample sizes at different phases. The current cohort description focuses on the longitudinal study (359 PLWH with >1 y follow-up) as shown in Table 1. However, it is hard to get the full picture based on mean values and ranges (e.g., mean time of follow-up, number of LP, and age at inclusion). For instance, there was a large variation in the “Number of lumbar punctures/individuals: 6 (range 2–30)”. It would help to include distribution plots or provide subgroup statistics within smaller ranges.2. As the authors presented, ART has an impact on CSF measures. Considering the wide ranges of follow-up time and number of LP, I assume some participants were recruited more recently than others. It would help to add “year of inclusion”. This information is also important to the future-plan assessment. For instance, whether less participants would agree to the repeated LP in more recent years?
--

	3. One option to show the full picture would be adding a figure with x-axis indicating year (1985 to 2022) and y-axis indicating participants (1 to 359), and marking all the follow-ups of all participants. A different color could be used to indicate participants eligible for future study. Minor 1. Put strengths before limitations in the section “Strengths and limitations”. The authors might need to consider the order of listed limitations, and whether it is necessary to list all these limitations. As I understand, the main limitation was the “Infrequent use of extensive neuropsychiatric testing” in this cohort. The other two limitations were more general (e.g., the difficulty to do yearly repeated LP due to Covid-19 pandemic) and could not be improved by changes in protocols. 2. As described in the Authors’ Guideline, it would be nice to have a section “Collaboration” for potential collaborators. “Authors should include a section on what data will be available, to whom, how it can be accessed and what restrictions to reuse may apply. (This should be in the text, not the data sharing statement.) Please also state what kind of collaboration you are encouraging.”
--	--

VERSION 1 – AUTHOR RESPONSE

Reviewer: 1

Dr. Beatriz Azevedo , Universidade Federal do Rio de Janeiro

Comments to the Author:

First of all, congratulations for the work developed by the authors, the compilation of information and findings are extremely important in the development of studies on HIV, the Central Nervous System and its correlation with antiretroviral therapies.

I only make one observation, I did not visualize the statistical part of the work, if any test or statistical program was carried out.

Statistical program has been included- Lars Hagberg

Reviewer: 2

Dr. Yuxin Shi, Fudan University, Shanghai Public Health Clinical Center

Comments to the Author:

The Gothenburg HIV CSF Study Cohort is a valuable cohort in studying the HIV effect on central nervous system (CNS). The authors have demonstrated the natural course of HIV infection and the impact of ART within CNS in their previous publications using data from this long-term follow-up study. I have a few suggestions.

Major

1. More details of the cohort or the subgroups are needed. Figure 1 provides a nice summary of the sample sizes at different phases. The current cohort description focuses on the longitudinal study (359 PLWH with >1 y follow-up) as shown in Table 1. However, it is hard to get the full picture based on mean values and ranges (e.g., mean time of follow-up, number of LP, and age at inclusion). For instance, there was a large variation in the “Number of lumbar punctures/individuals: 6 (range 2–30)”. It would help to include distribution plots or provide subgroup statistics within smaller ranges.

Table 2 has been included with time of inclusion, mean number of LP:s during each period, and participants eligible for further studies. Subgroups statistic is not possible because of large variation of the clinical outcome over time, but has been presented in several papers in the reference list.

2. As the authors presented, ART has an impact on CSF measures. Considering the wide ranges of follow-up time and number of LP, I assume some participants were recruited more recently than others. It would help to add “year of inclusion”. This information is also important to the future-plan assessment. For instance, whether less participants would agree to the repeated LP in more recent years?

Table 2 has been included with time of inclusion, mean number of LP:s during each period, and participants eligible for further studies.

3. One option to show the full picture would be adding a figure with x-axis indicating year (1985 to 2022) and y-axis indicating participants (1 to 359), and marking all the follow-ups of all participants. A different color could be used to indicate participants eligible for future study.

Such figure may not give more information than table 2 and will be very large

Minor

1. Put strengths before limitations in the section “Strengths and limitations”. The authors might need to consider the order of listed limitations, and whether it is necessary to list all these limitations. As I understand, the main limitation was the “Infrequent use of extensive neuropsychiatric testing” in this cohort. The other two limitations were more general (e.g., the difficulty to do yearly repeated LP due to Covid-19 pandemic) and could not be improved by changes in protocols.

Strengths have been placed before limitations according to the reviewer

2. As described in the Authors’ Guideline, it would be nice to have a section “Collaboration” for potential collaborators. “Authors should include a section on what data will be available, to whom, how it can be accessed and what restrictions to reuse may apply. (This should be in the text, not the data sharing statement.) Please also state what kind of collaboration you are encouraging.”

Collaboration” for potential collaborators has been included

1. Author Affiliations

Please remove the duplicate Author Affiliations just before the reference list.

Done

2. Abstract

Abstract in ScholarOne is different from main document. Please ensure that the Abstract in ScholarOne and main document are the same.

Done

3. Funding Statement

We have noticed there is a missing funder in ScholarOne as shown in the main document. Please ensure that the funding statement in ScholarOne and main document are the same.

Checked to be the same

4. Competing Interest

Competing Interest Statement in ScholarOne is different from main document. Please ensure that the Competing Interest Statement in ScholarOne and main document are the same.

Checked to be the same

5. Author Contributions

Author Contributions Statement in ScholarOne is different from main document. Please ensure that the Author Contributions Statement in ScholarOne and main document are the same.

Checked to be the same

6. Patient and Public Involvement

Has been included

. Tables

Tables in main document should be editable and in tabular format with rows and columns (Tables should have visible horizontal and vertical cell borders or gridlines).

Done

8. Figure/s

- Figure/s should not be embedded

Please remove the embedded figure in your main document. Please note that figure should be uploaded separately in ScholarOne.

Uploaded separately

9. Point by Point Response

Please provide a point-by-point response to the Editor's comments and reviewer's comments and submit them in Word Format.

Done

1. Author

Co-author 'Magnus Gisslén' is missing in the ScholarOne system. Kindly ensure that the arrangements of authors in your main document and Scholar One submission system are the same.

Checked to be the same

2. Author Contributions

- I have noticed that the initial 'MSG' is included in your Contributorship statement. However, checking the author list I cannot find a name with this initial. Kindly confirm. Please note that if this author does not fulfill all ICMJE guidelines for authorship, then this author should be moved to the acknowledgment section.

Checked

- I have noticed that the name 'Magnus Gisslén' is included in your author's list. However, upon checking the contributorship statement, I cannot find an initial that corresponds to its name. Kindly confirm.